# Learning to Co-Teach: A Systematic Review

**Anna Rytivaara [1]**, **Raisa Ahtiainen [2],***, **Iines Palmu [3]**, **Henri Pesonen [4]** and **Olli-Pekka Malinen [2]**

[1] Faculty of Education and Psychology, University of Jyväskylä, P.O. Box 35, 40014 Jyväskylä, Finland; aryti@jyu.fi
[2] Faculty of Educational Sciences, University of Helsinki, Siltavuorenpenger 5, P.O. Box 9, 00014 Helsinki, Finland; olli-pekka.malinen@helsinki.fi
[3] Valteri Centre for Learning and Consulting, Kukkumäentie 27, 40600 Jyväskylä, Finland; iines.palmu@valteri.fi
[4] Department of Special Needs Education, University of Oslo, Blindern, Postboks 1140, 0318 Oslo, Norway; henri.pesonen@isp.uio.no
* Correspondence: raisa.ahtiainen@helsinki.fi; Tel.: +358-29-4120-401

**Abstract:** Research on how teachers learn to co-teach is scarce. In this systematic review, the PRISMA method was used to examine the relationship between teacher learning and co-teaching in professional development programmes. Inclusion criteria was used to identify 567 articles on K–12 co-teaching, published in 2009–2018. A detailed analysis of nine articles revealed that the linkage between co-teaching and teacher learning remained narrow. Various programmes showed that the existing understanding of co-teaching or teacher learning was not used efficiently. Considerable variation in the programmes regarding the concepts, methods, and practices highlight the importance of conducting future research.

**Keywords:** co-teaching; teacher learning; professional learning; professional development

## 1. Introduction

Since the seminal article on co-teaching models by Friend and Cook [1], co-teaching has taken root in both classrooms and research. Co-teaching is widely examined at all levels of education from kindergarten to higher education, covering various subject areas and several research fields, such as coaching and co-teaching as a tool of teacher training (e.g., [2–4]). Co-teaching is generally defined as a collaborative practice in which two or more teachers plan, teach, and evaluate together as a group of learners (e.g., [5–8]). Moreover, as most of the literature on co-teaching draws from inclusive education aiming for high-quality education for all learners, it is defined particularly as a practice between a special education teacher and a general education teacher; yet, it can be practised between any two teachers (e.g., [9]). In our understanding, co-teaching is a multifaceted practice based on teachers' shared vision and responsibilities concerning teaching and learning for all students [5].

Much of the existing oeuvre of research has focused primarily on co-teaching models, and the trend has resulted in the prevailing understanding of the most common model in classrooms being the simplest one, that is, one teach–one assist [10,11]. However, while several reasons probably explain the situation, relatively little is known about how teachers learn to co-teach.

In this review, we investigated the relationship of co-teaching and teacher learning in more detail. We decided to limit the scope of our investigation to the literature focusing on co-teaching between at least two qualified teachers in K–12 education (K–12, from kindergarten to high school, refers to publicly supported education system in the US and is similar in many other countries) and chose to look at the studies reporting professional development (PD) programmes related to co-teaching. PD programmes were chosen as the focus of this review because PD inherently contains the premise of goal-oriented teacher learning aimed at changes in teachers' thinking and/or practice.

In discussing teacher learning as a focus of research, Kennedy [12] posed three main questions for researchers: first, what is it that teachers are supposed to be learning; second, what is the process of how teachers learn; and third, how can teacher learning be evaluated. These questions led to this review, as we applied them in the context of co-teaching and teachers' professional learning.

## 2. Learning within Co-Teaching

### 2.1. Features of Co-Teaching: Practices and Partnership

To understand what teacher's need to know in order to co-teach, co-teaching can be split into practical and relational features (Figure 1). While these features are not exclusive, studies on co-teaching often focus on one at a time. In the first strand of studies focusing on co-teaching practices, co-teaching is often described through a set of models indicating the way teachers take turns, interact, and engage with smaller student groups during the lessons [6]. A typical set of co-teaching models consists of the following: a model in which a teacher is teaching while one is observing the classroom, parallel teaching in which both teachers are teaching a separate group of students, station teaching in which students work in various "work stations", alternative teaching in which teachers alter the one who is leading the instruction, team teaching in which teachers work equally with all students but have flexible roles, and one in which one is teaching while one takes an assistant role. Of these, the last one seems to be the most frequently used [10,13,14]. The model's approach puts a lot of emphasis on classroom logistics and division of tasks between teachers [5,9]. Moreover, they construct a picture of models as the qualification to co-teaching, as if co-teaching was a simple tool a teacher can pull out of a toolbox and apply anytime in any classroom with any colleague. Especially in the inclusive setting, this approach is highly challenging as it leads special education teachers to be left with an assisting role while the general education teacher retains the decisive power during co-teaching [15].

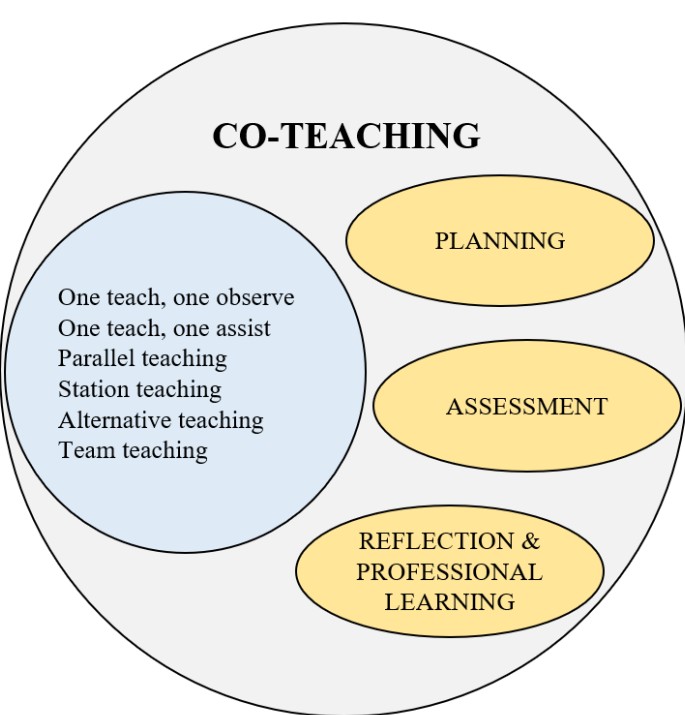

**Figure 1.** Features of co-teaching.

In the second strand of studies, co-teaching is approached as a partnership within which both teachers have equal responsibility over the lesson planning, instruction and assessment of the students. Within this approach, researchers have argued for a more delicate understanding of co-teaching and highlight the complexity of the formation and

development of well-functioning co-teaching partnerships [5,16,17]. These studies, often through long-term observation and teacher interviews, have explored how co-teaching partnerships are developed through negotiation and teachers sharing knowledge about their understandings of teaching and student learning [16–19].

The main distinctive feature between these strands might be the role of reflection. Fluijt and colleagues [5] argue for the importance of team reflection as a means to face and accept the complexity of the work, and as a path towards professional learning. By reflecting on their instructional roles and mutual interaction, teachers become more effective in their instruction and feel more satisfied with their co-teaching partnership [16]. Co-teachers' shared vision directs the goal setting for development in terms of both their students and the professionalism of their partnership [5]. Also, the trust between teachers develops through mutual care and respect for each other as professionals [16]. As a result of co-reflection and shared efforts, co-teaching partnership is built on teacher professionalism and teacher practice, and development of these two [5] rather than like-mindedness, friendship, or a compatibility between two teachers [9]. The challenge of co-teaching partnerships is that they take time and effort to develop, as both teachers need to be committed to the process of mutual learning, reflection, communication, and sharing [16,17]. However, as a result, knowledge of teachers with different backgrounds (e.g., special education, general education) can be used flexibly to benefit all students, rather than seeing the scope of teachers' practise as separate from each other [18].

### 2.2. Teacher Learning as Change in Teacher Thinking and Practices

In his discussion of the relationship between teaching and learning, Hirst [20] (p. 12) defined that "the end achievements of learning are new states of the person". His rather holistic view captures two essential—while not exclusive—elements of teacher learning, the learning process, and the aim of that process. In particular, the process can be elaborated as participation in a learning activity [21]. The aim of teacher learning is to change teachers' cognition and knowledge, beliefs, behaviour, skills or attitudes [22–24], or even teacher identity. Change in teacher knowledge can apply to their practical knowledge, which comprises experiential knowledge, formal knowledge and personal beliefs [25], or teachers' narrative knowledge composed of teachers' experiences over time, place, and relationships [26]. The role of the teacher identity appears to be playing a role in teachers' professional learning both as a target of the intended change and interacting with the learning process [27–29]. Teachers themselves are also individuals with variation in their willingness to learn new things [30].

### 2.3. Factors Promoting Teacher Learning

Following Kennedy's [12] second question, how teachers learn, teacher learning is generally understood as a change resulting from the involvement in learning activities [31]. We divided the features of effective professional development (PD) [32,33] (pp. 12–15) into two strands, structural features and features related to the actual learning process. Applied in co-teaching, the structural features cover the modelling of good co-teaching practices and design that allow teachers to connect their learning with their classroom practices. Teachers need to be involved in the programme for enough hours over a prolonged time span. Moreover, the programme design should be clearly linked with the learning goals, covering both the formal and the relational features of co-teaching, while also noticing teachers' existing practical knowledge. In addition, teachers sharing responsibility over the learning goals, content, and design of the PD can enhance their learning.

Teacher collaboration plays an important role in the features related to the learning process. Kwakman's [21] categorisation of teachers' learning activities into individual, instructional, and collaborative activities demonstrates this well, as nearly all activities can and are used collaboratively, even individual-level activities such as reading, experimenting, and reflecting. In fact, many instructional and collaborative activities mentioned in the earlier literature on teacher learning are very typical to co-teaching: co-planning,

choosing instructional activities and developing class materials together with colleagues, as well as sharing ideas and knowledge with colleagues [21,34–36]. Similarly, active and inquiry-based learning [32,33] may occur in ordinary co-teaching partnership [18]. Thus, PD programmes on co-teaching should support teachers learning together but also acknowledge their learning from each other.

Teachers learn the features of co-teaching through both formal and informal learning processes [31,37]. On one hand, formal learning opportunities appear to emphasise co-teaching practices covering topics of co-planning, knowledge about co-teaching models, and issues to consider and discuss together. On the other hand, the relational features of co-teaching, such as how to build trust and commitment with one's co-teaching partner, are often learnt informally in classrooms [16]. While the development of co-teaching partnership is generally acknowledged as an important issue in effective co-teaching [5], the relational part of co-teaching as a target for teacher learning is an understudied area. However, co-teaching per se appears to involve a fair number of learning opportunities as teachers share their beliefs, knowledge, and practical skills with their partner. Evidently, this has the potential to change their thinking as they negotiate and integrate their individual thinking and practices to develop joint co-teaching practices.

### 2.4. Evaluation of Teacher Learning

Kennedy's [12] third point is about the evaluation of teacher learning. Evaluation should always be in relation to the learning goals; what teachers are supposed to learn during PD. Kennedy [12] (pp. 152–153) has noted the discrepancy between teacher learning and student learning: While the ultimate goal of teachers' work, and thus also of teacher PD, is generally considered to be student learning, the linkage between teacher and student learning is far from straightforward. Thus, the frequent practice of evaluating teacher learning through student learning is highly problematic.

The means for evaluating teacher learning have to be connected with the understanding of and definition of learning in that specific context, such as a PD programme. Thus, direct means for evaluation can provide a relevant data source regarding changes in teachers' knowledge, beliefs, and behaviour. For example, changes in teacher behaviour can be examined through observation, or changes in teachers' thinking can be studied by interviewing teachers or using questionnaires. Teacher learning can also be evaluated by indirect means. By using multiple informants (e.g., teachers, students, researchers) and data sources (e.g., measures of student learning, teacher/student surveys, interviews), it is possible to obtain a multifaceted picture of the impact of PD programme on teachers (e.g., [31]).

Direct and indirect data sources can also be mixed. Researchers have used video-recall interviews and engaged with teachers in discussions around the themes arising from the materials collected through observations and interviews [31]. That is also a means to study the learning process in more detail; how it proceeds and how teachers make sense of their own learning. Finally, it is possible that even a year-long involvement in learning activities does not necessarily lead to changes in teachers' conceptions or behaviour [22].

### 2.5. The Purpose of This Review

The aim of this review is to explore the relationship between teacher learning and co-teaching in the context of professional development on co-teaching. We argue that teachers' learning process within co-teaching is a difficult phenomenon to recognise, and thus often goes unheeded. We will address the following research questions:

1. What features of co-teaching are the focus of teacher learning in the studies of professional development programmes on co-teaching?
2. How is teacher learning supported in the studies of professional development programmes on co-teaching?
3. How is teacher learning investigated/evaluated in the studies of professional development programmes on co-teaching?

## 3. Methods

### 3.1. Protocol

We used an evidence-based Transparent Reporting of Systematic Reviews and Meta-analysis (PRISMA) protocol as a guide for conducting this review [38]. This protocol has been used as a guiding principle in various review studies (e.g., [39]) to ensure the quality of review methods and reporting. The description of PRISMA protocol, including the accompanying checklist and flow diagram, is available online [40]. In the following, we outline how the steps were adapted in this systematic review.

### 3.2. Eligibility Criteria, Information Sources, and Search Strategy

This review is part of a larger review project examining co-teaching from several thematic perspectives. From the original research, covering all of the co-teaching studies of qualified teachers working in K–12 education, the focus of this review is on teachers' professional learning within co-teaching.

We searched in the two largest databases covering educational research, Ebsco and ProQuest. The criteria for including papers were: First, the abstract or the title had to include at least one of the following: "co-teaching" OR "coteaching" OR "co-teach" OR "coteach" OR "co-teach" OR "coteach" OR "co-taught" OR "cotaught", while not including any of the following: "higher education" OR "college" OR "university" OR "post secondary" OR "post-secondary" OR "postsecondary" OR "tertiary" OR "vocational". In both databases, we restricted the results to peer-reviewed articles published in English in scholarly or academic journals within the last ten years at the time of the search, 2009–2018. In sum, the criteria for inclusion were (1) the abstract or the title had to include one of the abovementioned keywords, (2) peer-reviewed, (3) published in English, (4) published in scholarly or academic journal, and (5) published in 2009–2018.

The Ebsco search resulted In 393 items, and 392 after the removal of a duplicate. The search in ProQuest resulted in 571 and 435 items, accordingly. The result after removing the doubles in the two databases was 567 articles.

### 3.3. Selection Process

The study's selection process is described in Figure 2. The screening of abstracts (n = 567) was conducted according to the inclusion/exclusion criteria described in Section 3.2. Each author also read five randomly selected full-text reports, after which the team confirmed the existing selection criteria. The screening of full-text reports included two rounds of assessment. In the first round, 174 reports were sought for retrieval. Full-text versions were found from 154 articles, which were assessed for eligibility. As a quality assurance measure, one-fifth of the reports were assessed by two authors independently. Data on the eligibility criteria were collected into an online spreadsheet which all authors of this review could access and edit. At the end of the first round, the authors convened to decide on the exclusion of the reports assessed as borderline cases. These borderline cases were, for example, reports that mentioned co-teaching, but the focus appeared to be on another topic and the research questions did not have direct connection to co-teaching. In total 98 full-text reports passed the first round of screening.

At the beginning of the second round of full-text screening, one inclusion/exclusion criterion, namely focus on co-teaching, was clarified so that co-teaching or an equivalent term (e.g., team teaching) had to be explicitly mentioned in the Introduction, Methods and Discussion sections of the report. In addition, studies in which co-teaching was only executed for the study, were excluded. The other inclusion/exclusion criteria remained similar to the first round. Each full-text report was read by a review author who had not read it in the first round. In total, 88 full-text reports passed the second round of screening.

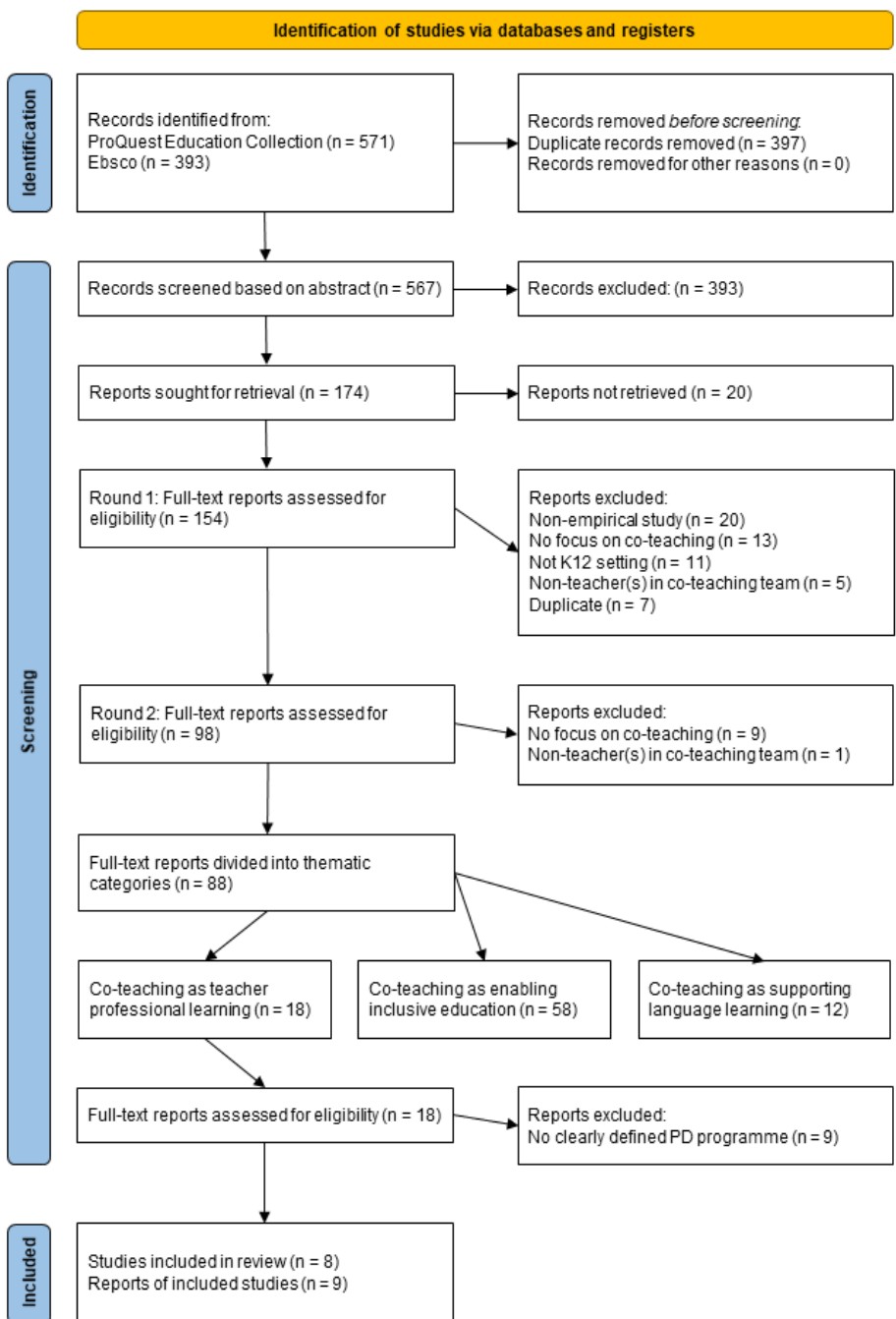

**Figure 2.** PRISMA flow diagram for the study selection process.

In the next phase of the study selection process, the full-text reports were thematised into three jointly agreed categories: (1) co-teaching as teacher professional learning (2) co-teaching as enabling inclusive education, and (3) co-teaching as supporting language learning. All of the review authors participated in this. In addition, reports assessed in this respect as borderline cases were discussed and decided collectively among the team. Reports falling into categories (2) co-teaching as enabling inclusive education, and (3) co-teaching as supporting language learning, were excluded from this review, leaving 18 full-text reports at the end of this selection phase.

The final phase of the study selection process was to exclude reports that did not focus on a clearly defined teacher professional development programme for improving co-teaching. Based on this criterion, nine reports were excluded. As two of the remaining

reports drew from the same study, the final sample comprised eight studies and nine full-text reports.

### 3.4. Data Collection Process and Data Items

The data from the nine reports (Table 1) were extracted to an online spreadsheet which was an extended version of the spreadsheet used during the study selection process. The two first authors of this review did most of the data extraction and negotiated agreement through discussion. The other authors followed the process by accessing the online spreadsheet and providing feedback when necessary.

**Table 1.** Main characteristics of the analysed studies in alphabetical order.

| Reference | Title | Country | Context | Study Design |
|---|---|---|---|---|
| Bryant-Davis, Dieker, Pearl, and Kirkpatrick (2012) | Planning in the Middle: Co-Planning Between General and Special Education. | USA | Middle school. | Qualitative |
| Faraclas (2018) | A Professional Development Training Model for Improving Co-Teaching Performance. | USA | 3 middle schools, 3 high schools from 4 school districts. | Experimental AND/OR a randomised pretest-posttest control group design |
| Jang 2010 | The Impact on Incorporating Collaborative Concept Mapping with Coteaching Techniques in Elementary Science Classes | Taiwan | Elementary school. | Mixed-method, quasi-experimental |
| Nilsson, P. (2015) | Catching the moments— coteaching to stimulate science in the preschool context. | Sweden | Preschool. | Qualitative, interview data |
| Pearl, Dieker, and Kirkpatrick (2012) | A five-year retrospective on the Arkansas Department of Education Co-teaching Project | USA | 208 elementary, middle and high schools in 143 school districts. | Mixed-method |
| Ploessl and Rock 2014 | eCoaching: The Effects on Co- Teachers' Planning and Instruction. | USA | Elementary schools. | Single-case withdrawal design |
| Scheeler, Congdon, and Stansbery (2010) | Providing Immediate Feedback to Co-Teachers Through Bug-in-Ear Technology: An Effective Method of Peer Coaching in Inclusion Classrooms. | USA | Elementary and middle schools. | Multiple-baseline, across-participants design |
| Shaffer and Thomas-Brown (2015). | Enhancing Teacher Competency through Co-Teaching and Embedded Professional Development | USA | Secondary school. | Qualitative, interview data |
| Thomas-Brown and Sepetys (2011) | A Veteran Special Education Teacher and a General Education Social Studies Teacher Model Co-Teaching: The CoPD Model | USA | Secondary school. | Qualitative, interview data |

The analysis of the papers covered three themes: (1) the PD programme as the context of learning, (2) features related to the teacher learning process, (3) evaluation of teacher learning. In detail, the following items were extracted:

- PD programme characteristics (e.g., length, content and aims);
- intensity and timespan of co-teaching;
- teacher roles in co-teaching team;

- research questions;
- definition of co-teaching;
- co-teaching activities;
- justification for introducing co-teaching;
- co-teacher and student characteristics;
- study context (e.g., country, region, grade level);
- co-teaching implementation time span.;
- recognition of teachers' previous practical knowledge;
- description of teacher learning process;
- teachers' reported learning;
- learning activities;
- means of evaluating teacher learning.

*3.5. Study Risk of Bias Assessment*

To ensure the trustworthiness of the review process, at the beginning, each author read all of the 567 article titles and abstracts to check that all material was relevant for the larger review research project. This was followed by each author reading five randomly selected articles that were discussed until agreement on the sought data items were reached. This was followed by coding the data items and placing them into the Excel sheet. Next, research questions concerning this review were utilised in further coding the material. In each full-text paper-reading round, each paper was read by at least two members of the five-person research team. Moreover, in each full-text paper reading round, each research team member was appointed a different set of papers to read. The items and notes in the Excel sheet were cross-checked after each reading round and discussed among the research team several times during independent reading and in the data validation meetings that involved the whole team. In the meetings, any contradictions or border-line cases were discussed until a consensus decision was reached. Furthermore, the authors of this article—who also conducted the systematic review—are experienced researchers with extensive expertise in the topic and the applied methods. We also used peer-debriefing for the purposes of trustworthiness [41] during the research process to discuss the findings with experienced peers (e.g., colleagues during conference presentations) that had no personal connection to the project.

**4. Results**

The main characteristics of the PD programmes are described in Table 2.

**Table 2.** The main characteristics of the PD programmes.

| Reference | Programme Goals | Programme Duration | PD Programme Description | Participants | Conclusions/Main Findings |
|---|---|---|---|---|---|
| Bryant Davis et al., 2012 | To provide professional development in the area of co-teaching and to analyse current practices. | School year | One-year programme included one-day training session for teachers, onsite visits upon request and monthly webinars. Web page and email/telephone support as needed. Four check points for lesson plans. | Special education and regular education teachers | Teacher learning reported in another publication from the same project (see below Pearl et al., 2012). |
| Faraclas et al., 2018 | Training was designed to foster parity between co-teaching partners, and to provide teachers with research-based strategies to effectively instruct students with disabilities. | Eight weeks | Programme included six two-hour training sessions on five areas of co-teaching performance: planning, instruction, classroom management, behaviour management, assessment. Observation of teachers. Feedback. | 48 special and regular education teachers in 24 co-teaching dyads | Participation in PD resulted in teachers using more approaches and in a more equal relationship in the classroom. |
| Jang 2010 | To integrate a collaborative concept-mapping technique into co-teaching. | Eight weeks | Teachers co-taught a 40-min lesson thrice a week. Programme included after-class discussions, planning and evaluation. Teachers met weekly. | 2 science teachers | In interviews, both teachers experienced having learned from the programme. |
| Nilsson 2015 | To increase preschool teachers' knowledge of specific domains of science and to help them reflect on how such ideas, through co-teaching, can be developed and integrated into their own practice. | School year | Programme included $4 \times 3$ h university lectures and hand-on activities with teams from three preschools, two video-recorded sessions in preschool, and analysis of videos with researchers. Teachers read science literature and met regularly with other teams to discuss science teaching. | 9 preschool teachers | Change was reported regarding three areas: experiencing confidence, enthusiasm and trust; shared responsibility leading to new ways of approaching science; engaging each other and the children in collaborative discussions. |
| Pearl et al., 2012 | To create a statewide system for co-teaching and to provide professional development for co-teaching partnerships | School year | One-year programme included one-day training session and one-day follow-up session for teachers, $5 \times 5$ h webinars, two half-day onsite coaching visits, and individualised annual reports for schools. | 789 special and general education teachers | All teams reported positive change. |
| Ploessl and Rock 2014 | To support teachers' planning and implementing varied co-teaching models, use of student-specific accommodations and modifications, and positive behavioural interventions and supports through eCoaching | Not available | Teachers were given feedback through bug-in-the ear device for four 30-min sessions (i.e., planning-teaching-planning-teaching), and this was repeated after a withdrawal phase. | 6 special and general education teachers in 3 co-teaching dyads | All teams increased the relationship between the co-teaching models planned and those implemented in classroom. |

**Table 2.** *Cont.*

| Reference | Programme Goals | Programme Duration | PD Programme Description | Participants | Conclusions/Main Findings |
|---|---|---|---|---|---|
| Scheeler et al., 2010 | To help teachers with co-teaching [completion of three-term contingency trials] | Three months | In the three-month programme teachers gave immediate feedback for their co-teacher through bug-in-the ear device. Teachers co-teach several lessons weekly. | 3 special education teachers and 2 math and 1 language arts teacher in 3 co-teaching dyads | All teams improved in providing feedback to their co-teaching partners. |
| Shaffer and Thomas-Brown 2015 | A veteran special education teacher to provide daily professional development training to a general education teacher. | One semester | Daily co-teaching for 50-min lesson. Programme included meetings and debriefing at the end of each day. | 1 special education and 2 social studies teachers | In interviews, teachers reported change in their thinking and pedagogical practices. |
| Thomas-Brown and Sepetys 2011 | A veteran special education teacher to provide daily professional development training to a general education teacher. | One semester | Daily co-teaching for 50-min lesson. Programme included meetings and debriefing at the end of each day. | 1 special education and 2 social studies teachers | In interviews, teachers reported change in their thinking and pedagogical practices. |

*4.1. Focus of Teacher Learning*

4.1.1. Features of Co-Teaching

This section is based on the introduction, the literature review and the method sections of the selected reports and describes how the co-teaching framework was set in terms of features of co-teaching as practice and as a partnership.

In seven studies [42–48], co-teaching was the practice of a special and a general education teacher that supports the implementation of special education services in mainstream classroom settings. In the remaining two studies [49,50], co-teaching was practiced by two or more teachers in mainstream settings, aiming to enrich teaching methods and student learning. In total, three studies explicitly mentioned co-teaching taking place in one shared space [42,44,45]. A set of co-teaching models was included in seven articles [42–49]. All of the studies either mentioned [42–48] or described in more detail [43–50] a variety of 'co-practices' (e.g., co-planning, co-instruction).

In terms of co-teaching as a partnership, three studies [43,49,50] focused more on collective responsibilities regarding co-teaching, student learning, and planning instructions for all students. Faraclas and colleagues [43] explicitly mentioned teachers' equal participation in teaching. Jang [49] noted that co-teaching includes sharing feelings and thoughts related to the interaction between teachers and with students. Moreover, Faraclas et al. [43], Jang [49], and Thomas-Brown and Sepetys [48] acknowledged the importance of skills for joint reflection of challenging situations emerging in co-teaching partnerships. Further, Scheeler and colleagues [46] pointed out the meaning of mutual feedback and support in the development of co-teaching partnerships. Two studies [47,48] argued that to co-teach, teachers ought to discuss their expectations for co-teaching and their roles as teachers in the co-teaching partnership.

The definitions of co-teaching included clear aspects of the teaching profession and teacher learning in four studies [47–50]. Through working in co-teacher partnerships and mutual reflection, teachers learn alternative ways to teach [49], other teachers' perspectives on teaching [47,48], and in general, teachers learn about teaching [48]. Overall, six studies focusing on inclusion [42,43,45–48] referred to the various kinds of professional knowledge of special and general education teachers. The former are experts in supporting students, whereas the latter specialize in subject content knowledge. Furthermore, Jang [49] describes co-teaching as professional sharing. In total, two studies [45,48] explicitly acknowledge

that co-teaching requires the learning and training of new skills and knowledge. Only one study [45] noted that in co-teaching, teachers take new roles as teachers. In total, two studies [49,50] connected co-teaching to the socio-constructive model of learning and the zone of proximal development.

### 4.1.2. The Relation of Co-Teaching to Teachers' Learning

In the PD programmes we studied, the role of co-teaching was approached as a focus of teacher learning as well as a learning context of teacher learning. Overall, four PD programmes focused on improving co-teaching practice. In particular, they focused on co-teaching performance [43], co-teaching implementation and professional development [44], classroom application of co-teaching and to provide support to teams in how to plan effective lessons [42], and on the use of co-teaching models [45].

In total, five studies regarded co-teaching as a learning context. In Nilsson [50], teachers enhanced their science teaching skills through co-planning, co-teaching, and co-reflection within their team and other teams. Jang [49] described her study as having two foci: teachers learning to use concept maps as an instructional tool, and co-teaching. In Scheeler et al. [46], teachers were teaching each other to give feedback to students during a co-taught lesson by using ear-bug technology. In the studies by Thomas-Brown and Sepetys [48] and Shaffer and Thomas-Brown [47], co-teaching was used as a context where the subject-matter teacher would learn differentiation by observing an experienced special education teacher at work.

### 4.2. Support for Teacher Learning
#### 4.2.1. Duration and Intensity

The duration of the programmes varied tremendously from a year [42,44,50] or a semester [46–48] to eight weeks [49]. Ploessl and Rock [45] do not report the length of their programme. Similarly, the intensity of co-teaching varied a lot. In the four studies that reported it, the frequency of co-teaching varied from one subject [46] or three times a week [49] to daily co-teaching [47,48].

In total, four programmes combined on-going co-teaching with some external PD-training. The year-long programmes [42,43,50] (1) covered an intensive package of formal training, practical experimenting, on-site coaching, and webinars. Likewise, the PD-training reported by Faraclas [43] covered six two-hour training sessions within the eight-week period.

#### 4.2.2. Recognition of Teachers' Prior Knowledge

None of the studies reported using the teachers' previous knowledge or skills in planning the programme. However, teachers had opportunities to use their existing knowledge and skills in developing suitable examples of short phrases for giving feedback [46], applying collaborative learning approaches in the classroom [49], or in writing lesson plans [42]. Researchers also used teachers' co-teaching experiences in developing webinars [44] as part of the programme. Teachers' active learning and experimenting in co-teaching teams were reported in only one study [50].

#### 4.2.3. Learning Activities

Regarding the learning activities within the PD programmes, we divided the studies into three groups. The first group comprised studies with a diverse collection of learning activities in and out of the classroom [34,42,44,50]. For example, all four studies combined formal training sessions on co-teaching with working in small groups and used classroom observations with feedback. In Bryant Davis et al., Nilsson, and Pearl et al., teachers were also provided with reading from the literature. Additionally, Nilsson reports co-reflection and experimenting in the university's science laboratory as learning activities.

In the second group of studies, the main learning activity was co-teaching, while not providing teachers with external support to learn from their co-teaching experiences. Thus,

teachers' learning was based on their co-teaching activities, observing their partner, and after-class discussions [47–49].

The third group of studies comprised two clearly defined intervention studies, one in which teachers were supporting each other on giving feedback to students [46] and one in which the focus was on learning about different co-teaching models [45]. In Ploessl and Rock, the dyads were supported through eCoaching during their co-planning and co-teaching.

To sum up, in most cases the teachers were learning together and learning from each other, or from an outside trainer. In three studies, it is difficult to name whether the main source was the training, learning together, learning from each other, or all of these [42–44].

### 4.3. Evaluation of Teacher Learning

In three studies [43,45,46], researchers observed the teachers studying, to changes in their behaviour. Pearl et al. [44] asked teachers to fill in pre- and post-measure questionnaires on the effectiveness of co-teaching. The 38-item Colorado Assessment of Co-teaching tool scored on three factors: personal prerequisites (15 items), professional relationship (9 items) and classroom dynamics (14 items). In two studies [47,48], teacher learning was indicated by the teachers' own description of the changes they had made in classrooms. Nilsson [50] analysed changes in teachers' thinking from recorded video-club meetings and interviews. In Jang's [49] study, change was studied by comparing students' pre- and post exam results, but it also emerged in teachers' comments/interviews.

In one study [49], change was not evaluated at all, and in another study [45] the researchers reported that one out of three pairs increased the number of co-teaching models they used, and that all dyads became better at following their plans about what models were they were about to use. In another seven studies, some level of change occurred.

## 5. Discussion

The aim of this review was to examine the relationship between teacher learning and co-teaching in the context of PD programmes. Our findings offer a novel contribution to the co-teaching literature, by highlighting fine-grained interfaces between teacher learning, co-teaching, and PD programmes, which have previously remained unrecognised. We will discuss the findings from two main perspectives: First, based on the findings about the focus of teacher learning and the features of co-teaching, we will discuss how the findings reflect different conceptualisations of co-teaching. The second perspective, what conceptualisation of teacher learning do the findings reflect, draws from the findings regarding the support for learning and evaluation of teacher learning. These perspectives are intertwined, as the conceptualisation of co-teaching behind PD programmes has implications for what knowledge and skills teachers are supposed to acquire in the programmes, and how their individual learning is connected to the learning of their co-teaching team.

Analysis on the focus of teacher learning revealed that co-teaching was described in the background section of the studies rather differently from how co-teaching was understood in the actual PD programmes. In particular, teachers' joint learning and partnership aspect of co-teaching were generally noticed in the background section but not considered as part of the structure of the PD programme or in the findings. Co-teaching appeared to provide context for teachers' learning, but the aim of learning was on teaching the participants separate features, some of which were only loosely connected to co-teaching, and none of which were aimed at developing teachers' co-teaching partnerships or reflection skills. This reflects quite a narrow understanding of co-teaching as a set of technical skills and practices rather than as a joint effort of two or more teachers with common goals to support the learning of all students in their classrooms. Accordingly, all but one study [50] approached teacher learning within co-teaching as an individual-level phenomenon, instead of understanding co-teaching as a team effort; a kind of an extra member of the team in addition to the individual teachers.

Another issue was the short timeframe of some programmes. In what is perhaps a chicken or egg situation, we wonder whether a short timeframe is related to the idea of co-teaching as practices which are simple and fast to acquire. Another perspective could be that PD programmes need to be kept short and learning goals concrete to make it easier for teachers to participate. Either way, co-teaching is a long-term process which requires time, and such programmes may maintain a false idea of co-teaching as a toolbox that can be used anytime, with any colleague and in any student group. Nevertheless, it would be important to pay attention to teachers' reflection skills, which would support their learning from practice to becoming conceptualised and thus helping them to use their learning in new settings [51]. Reflection skills would also support teachers in building co-teaching partnerships with their colleagues [5,52]. Two studies [45,46] gave hints about how teachers might learn together, even if the learning was not aimed at enhancing co-teaching teams but individual teachers.

As co-teachers in general, as well as in the studied PD programmes, evidently tend to learn together; this informal and collective learning goes unnoticed from the researchers who tend to focus on individual-level formal learning of the programme content. Our findings, especially regarding the second and third research questions, raise concerns about such misdirection of attention, which emerge despite the investigated PD programmes are built around co-teaching, which, for one, is inherently based on teacher collaboration and interaction. This misdirection results in a kind of discrepancy in teachers learning together and from each other, while their collective learning is rarely noticed or acknowledged by the programme organisers. This suggests that powerful learning opportunities may go unheeded, and the informal forms of learning are left unsupported.

Accordingly, most studies in this review were based on the idea of teacher learning as something observable in the short-term, such as a new practice or obvious change in one's behaviour. It is essential to question how permanent these changes actually are in the context of co-teaching, considering the reality in schools which makes teachers change classrooms and co-teachers frequently. In addition to not getting information about teachers' learning in the long term, such an approach to teacher learning ignores other types of learning, such as transformative learning or changes in teachers' professional identities [29,53].

Co-teaching draws strongly from the framework of inclusive education, which has resulted in co-teacher partnerships most frequently being composed of a general education and a special education teacher [15]. This was also the case in the studies in our review. Nevertheless, even though teachers' existing practical knowledge is an essential part of developing a co-teaching partnership and in their professional learning [17], none of the programmes addressed the participating teachers' previous knowledge and skills. However, in two studies [47,48], co-teachers were assumed to have different knowledge and skills due to their different training as teachers. What was noteworthy in these cases was that only the subject-matter teacher was to learn from the special education teacher, rather than using the opportunity for teachers sharing in their practical knowledge and both learning from each other. In the context of teacher learning, instead of such a presumed categorisation of one being labelled as a content expert and the other as an expert in differentiating instruction, more effort needs to be put on the idea of both teachers learning from each other. This would make a fruitful starting point for a professional development programme.

According to previous studies, teacher learning in co-teaching is informal, job-embedded learning through which teachers appear to learn by doing; they learn by co-teaching and engaging in their co-teaching duties, which in turn result in more developed co-teaching practices and a mutual partnership with their co-teaching partner [17]. This involves teachers learning together as well as learning from each other, these two processes often occurring inseparably. In the studies included in this review, this aspect of teachers' learning was not noticed in most cases, and thus, not evaluated. However, a feature of co-teaching is that teachers learn from each other while working together, and thus these learning opportunities could well be enhanced in PD programmes. Making this implicit

learning explicit and visible could also be a means to enhance teachers' learning after and outside the programmes.

We need more studies on the learning of co-teaching teams, exploring the learning of teams and the individual teachers, and how these two are linked. The studies in this review failed to recognise co-teaching teams as units in which an individual teacher's learning is intertwined with teachers' collaborative learning as a team. Regardless of this, teachers' joint learning emerged in teachers' own description of the PD programme. What was particularly noteworthy was that the teachers' experiences were reported even when the researchers did not discuss this informal learning explicitly.

*5.1. Limitations*

This review drew from a relatively small number of studies, yet the studies were carefully selected from an extensive body of co-teaching research in an on-going collaboration among the authors. While acknowledging that teacher learning is more than the sum of its parts, the small number of studies allowed us to analyse the PD programmes and the related teacher learning in detail. All participating teachers having volunteered to co-teaching is a possible limitation which could overemphasise their learning results. However, as we focused on analysing learning processes rather than learning results, this probably had no major role in our findings.

Some general dilemmas prevailed throughout the studies. First, in some cases teacher learning was mentioned very briefly as if it was a serendipitous notion more than a focus of professional development programme. Related to this, learning was not always specified; what was learned, or for example, the change in behaviour or in thinking was not discussed further, even if behaviour was the focus of learning. Another point for critical discussion might be the issue of whose perspective of learning was addressed: the researcher's or the teacher's. Nevertheless, as mentioned above, the focus of this review was on the learning processes and the relationships between co-teaching and teacher learning, rather than the results of the programmes.

*5.2. Conclusions*

The field of co-teaching appears to be mainly studied in the framework of inclusive education and has remained separate from the research field of teaching and instruction, and thus, teacher learning. This review is one effort to link these two. Our findings suggest that the relationship between co-teaching and teacher learning remained rather light in general. This is an important finding as teacher learning is a process in which the focus of learning, the means of learning, and the evaluation of learning are all interconnected. Thus, the conceptualisation of co-teaching affects what teachers are supposed to learn, and what they are supposed to learn should be inevitably linked to the learning methods. Moreover, the evaluation of a teacher's learning should focus on the learning goals of the programme. Our review also revealed that the literature on professional development programmes related to co-teaching varies regarding the concepts and methods as well as the actual co-teaching practices. This makes it challenging to draw reliable conclusions about the impact of such programmes on teacher learning.

In future studies, a microanalytical approach to co-teachers' communication during classes, breaks, and planning sessions would provide detailed information about the process of their on-going professional learning. This would further increase our understanding of the factors promoting and hindering effective job-embedded teacher learning. This setting might also let researchers distinguish the process of learning from each other and from the process of learning together and shed light on the relationship between the two. Moreover, when planning for professional development programmes on co-teaching, a deeper understanding of teacher learning as well as co-teaching might enhance teachers' joint knowledge construction.

**Author Contributions:** Conceptualization, A.R. and R.A.; methodology, A.R., R.A., I.P., H.P. and O.-P.M.; validation, A.R., R.A., I.P., H.P. and O.-P.M.; formal analysis, A.R. and R.A.; investigation, A.R., R.A., I.P., H.P. and O.-P.M.; data curation, A.R. and R.A.; writing—original draft preparation, A.R., R.A., I.P., H.P. and O.-P.M.; writing—review and editing, A.R., R.A., I.P., H.P. and O.-P.M.; visualization, I.P. and O.-P.M.; project administration, R.A. and R.A. All authors have read and agreed to the published version of the manuscript.

**Funding:** This research received no external funding.

**Data Availability Statement:** All reports included to this review are listed in the references [42–50] and they available on the respective journals' websites.

**Acknowledgments:** Open access funding provided by University of Helsinki.

**Conflicts of Interest:** This research did not receive any specific grant from funding agencies in the public, commercial, or not-for-profit sectors.

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
