# Peer review of "Learning to Co-Teach: A Systematic Review"

_education, doi:10.3390/educsci14010113_

Round 1
Reviewer 1 Report
Comments and Suggestions for Authors
This is a very well-researched and thoroughly performed article. Thank you so much for your contribution to scholarship and for such a thorough exploration of teacher co-teaching models and their relationship to teacher preparation programs. This is very thoroughly done; my one suggestion for improvement would be to include a diagram that summarizes the table of papers, but this is a minor addition.
Author Response
|
This is a very well-researched and thoroughly performed article. Thank you so much for your contribution to scholarship and for such a thorough exploration of teacher co-teaching models and their relationship to teacher preparation programs. This is very thoroughly done; my one suggestion for improvement would be to include a diagram that summarizes the table of papers, but this s a minor addition.
|
Thank you for your constructive and encouraging feedback. We have followed the PRISMA reporting guidelines for systematic reviews, and according to the guidelines, it is not common to use diagrams in summarizing the contents of the table consisting of selected papers (PRISMA, 2024, http://prisma-statement.org/ ).
|
Reviewer 2 Report
Comments and Suggestions for Authors
Paper seems to be adequately researched and presented. There are some questions:
1. How did authors "extract data"? What is that method? Was there coding? Memoing? If so, how? Was there a horizontalism? Not having a reliable/valid qualitative methodology creates difficulty for the readers. Also, what was "negotiated," and how were differences handled? There are both pros and cons for consensus.
2. The results could be improved with a visual and a better organization by theme. The authors present a lot of results, and while they are well done (including outliers/stubs), presenting a central theme under each with dimensions would be much easier to read and conceptualize as a whole.
3. References need cleaned up. Articles should not appear 2-3 times but rather only once. For example, Friend et al. is cited as article 6 and article Rytivaara, A., & Kershner, R. (2012) is cited as 25 & 36. Fluijt, D., Bakker, C., & Struyf, E. (2016) is 5, 10, & 18. Duplicates are unnecessary and make the paper more difficult to follow; MDPI usually finds such mistakes later.
3. Since the authors limited themselves to two databases that require submissions or partnerships, many research studies can be overlooked. When conducting PRISMA studies, using generic search engines, such as Google Scholar, etc., can often find overlooked items. A limitation seems to be the unlikely scenario there were only 9 articles. Did the authors consider the following articles:
Pre-service (outside your scope but relevant)
Weilbacher, G., & Tilford, K. (2015). Co-teaching in a year-long professional development school. School-University Partnerships, 8(1), 37-48.
Author Response
Education Sciences – responses to reviewers
|
Reviewer# 2 Comments and suggestions |
Response and actions taken |
|
1. How did authors "extract data"? What is that method? Was there coding? Memoing? If so, how? Was there a horizontalism? Not having a reliable/valid qualitative methodology creates difficulty for the readers. Also, what was "negotiated," and how were differences handled? There are both pros and cons for consensus |
In the review process we followed guidelines of the the PRISMA statement (PRISMA, 2024, http://prisma-statement.org/ ) which is one of the most frequently used guidelines for carrying out a review study - the inclusion/exclusion criteria are described in section 3.2., and we included studies that, e.g., covered: co-teaching studies of qualified teachers working in K-12 education, teachers’ professional learning within co-teaching, empirical study. - The screening of full-text reports included two rounds of assessment. - Each author also read five randomly selected full-text reports, after which the team confirmed the existing selection criteria throughout a consensus negotiation (only full-agreement accepted) The negotiation was about going through the criteria and reading the articles together, and based on that discussing whether articles met the criteria or not. Hence, it was about making sure that all researchers participating in the process had a shared understanding about the framework within which the work was done. |
|
2. The results could be improved with a visual and a better organization by theme. The authors present a lot of results, and while they are well done (including outliers/stubs), presenting a central theme under each with dimensions would be much easier to read and conceptualize as a whole. |
We would like to thank the reviewer for this suggestion. However, in the reporting of the results, we have followed the PRISMA statement guidelines, and according to the guidelines, it is not common practice to use figures or equivalent visuals when presenting the findings (PRISMA, 2024, http://prisma-statement.org/ ).
|
|
3. References need cleaned up. Articles should not appear 2-3 times but rather only once. For example, Friend et al. is cited as article 6 and article Rytivaara, A., & Kershner, R. (2012) is cited as 25 & 36. Fluijt, D., Bakker, C., & Struyf, E. (2016) is 5, 10, & 18. Duplicates are unnecessary and make the paper more difficult to follow; MDPI usually finds such mistakes later. |
We have cleaned up the references. |
|
4. Since the authors limited themselves to two databases that require submissions or partnerships, many research studies can be overlooked. When conducting PRISMA studies, using generic search engines, such as Google Scholar, etc., can often find overlooked items. A limitation seems to be the unlikely scenario there were only 9 articles. Did the authors consider the following articles: Pre-service (outside your scope but relevant) Bashan, B., & Holsblat, R. (2012). Co-teaching through modeling processes: Professional development of students and instructors in a teacher training program. Mentoring & Tutoring: Partnership in Learning, 20(2), 207-226. Seems to be on point / did you exclude?
Pancsofar, N., & Petroff, J. G. (2013). Professional development experiences in co-teaching: Associations with teacher confidence, interests, and attitudes. Teacher Education and Special Education, 36(2), 83-96.
Miller, C., & Oh, K. (2013). The effects of professional development on co-teaching for special and general education teachers and students. Journal of Special Education Apprenticeship, 2(1), n1.
Gallo-Fox, J., & Scantlebury, K. (2016). Coteaching as professional development for cooperating teachers. Teaching and Teacher Education, 60, 191-202.
Review articles of interest: Ghazzoul, N. (2018). Collaboration and co-teaching: Professional models for promoting authentic engagement and responsive teaching. Pertanika Journal of Social Sciences & Humanities, 26(3).
King-Sears, M. E., Stefanidis, A., Berkeley, S., & Strogilos, V. (2021). Does co-teaching improve academic achievement for students with disabilities? A meta-analysis. Educational Research Review, 34, 100405. |
Thank you for this comment. Referring to inclusion/exclusion criteria explained under the point 1 on this table the suggested articles did not meet the criteria: The Weilbacher & Tilford is not included, as it did not meet our criteria for qualified teachers.
Bashan, B., & Holsblat is not included, as it did not meet our criteria for qualified teachers.
Gallo-Fox, J., & Scantlebury, K. (2016), as it did not meet our criteria for qualified teachers.
Pancsofar, N., & Petroff, J. G. (2013) was not included as it was based on questionnaire data and the study did not include any form of professional development.
Miller, C., & Oh, K. (2013). This paper was not included because we excluded studies in which co-teaching was only executed for the study. This criteria was clarified in the revised paper in section 3.3.
|
Reviewer 3 Report
Comments and Suggestions for Authors
Dear authors,
you have done a good work. The topic is very interesting to be addressed in teachers training. It calls my attention that the term Co-teaching is not included in Eric thesaurus though I have found 69.000 references of co-teaching in Google academic.
The application of Prismas is correct though I have a question: in the third stage you decided to divide the papers into three groups. Finally, you decide to exclude inclusion and language to focus on the first one. Why three groups and not one since the very first moment? This is somewhat confusing to me.
Who are the authors who decide which report is adequate? What is their expertise? How were they selected? Are the “authors” the authors of this article? If so, can this compromise the reliability of the study? Sometimes we need experts out of the research to give us another perspective.
Only “9 articles” is a very poor sample to make general conclusions on any topic. And this is the weak point of the work. It is a pity since the topic is relevant and you have done a good job.
I would suggest going over the questions for future works and considering external experts to have other opinions.
Author Response
|
Reviewer# 3 Comments and suggestions |
Response and actions taken |
|
The application of Prismas is correct though I have a question: in the third stage you decided to divide the papers into three groups. Finally, you decide to exclude inclusion and language to focus on the first one. Why three groups and not one since the very first moment? This is somewhat confusing to me.
|
As we stated in the paper, “This review is part of a larger review project examining co-teaching from several thematic perspectives. From the original research, covering all co-teaching studies of qualified teachers working in K-12 education, the focus of this review is on teachers’ professional learning within co-teaching.” This is a substudy of the larger project. |
|
Who are the authors who decide which report is adequate? What is their expertise? How were they selected? Are the “authors” the authors of this article? If so, can this compromise the reliability of the study? Sometimes we need experts out of the research to give us another perspective. |
Thank you for this important feedback. We have added information about authors’ expertise (the researchers of the review) in carrying out these types of studies. We have written about seeking outside experts’ perspectives. The added information can be found under 3.5. Study risk of bias assessment in red font.
|
|
Only “9 articles” is a very poor sample to make general conclusions on any topic. And this is the weak point of the work. It is a pity since the topic is relevant and you have done a good job.
I would suggest going over the questions for future works and considering external experts to have other opinions.
|
Thank you for this feedback. We agree that nine articles is a small amount of studies and thus we definitely need more work on the topic.
We have included information about external experts in the 3.5. section about study bias (in red). In this section, we discuss peer-debriefing that was used for the purposes of trustworthiness. |